# Unplanned Return to the Operating Room after Elective Oncologic Thoracic Surgery: A Further Quality Indicator in Surgical Oncology

**DOI:** 10.3390/cancers14092064

**Published:** 2022-04-20

**Authors:** Francesco Petrella, Monica Casiraghi, Davide Radice, Claudia Bardoni, Andrea Cara, Shehab Mohamed, Daniele Sances, Lorenzo Spaggiari

**Affiliations:** 1Department of Thoracic Surgery, IRCCS European Institute of Oncology, 20141 Milan, Italy; monica.casiraghi@ieo.it (M.C.); claudia.bardoni@ieo.it (C.B.); andrea.cara@ieo.it (A.C.); mohamed.shehab@ieo.it (S.M.); lorenzo.spaggiari@ieo.it (L.S.); 2Department of Oncology and Hemato-Oncology, Università degli Studi di Milano, 20122 Milan, Italy; 3Division of Epidemiology and Biostatistics, IRCCS European Institute of Oncology, 20141 Milan, Italy; davide.radice@ieo.it; 4Division of Anesthesiology, IRCCS European Institute of Oncology, 20141 Milan, Italy; daniele.sances@ieo.it

**Keywords:** unplanned return, operating room, adverse events, complications, morbidity, mortality, thoracic oncology

## Abstract

**Simple Summary:**

An unplanned return to the operating room (UROR) is defined as a readmission to the operating room because of a complication or an untoward outcome related to the initial surgery. It has been widely used as an indicator of surgical care quality among hospitals. The aim of this paper is to evaluate the role of URORs after elective oncologic thoracic surgery in a high-volume, oncologic referral center, focusing on risk factors and variables that influence the UROR rate. Our findings disclosed that UROR is an effective and reliable quality indicator in oncologic thoracic surgical care; patients experiencing UROR after elective oncologic thoracic surgery have a higher morbidity and mortality rate when compared to standard thoracic surgery. Patients presenting complications after UROR had been submitted to a significantly longer first procedure, had a significantly longer length of stay and a higher post-operative mortality. Bronchopleural fistula remains the most lethal complication in patients undergoing UROR.

**Abstract:**

Background: An unplanned return to the operating room (UROR) is defined as a readmission to the operating room because of a complication or an untoward outcome related to the initial surgery. The aim of the present report is to evaluate the role of URORs after elective oncologic thoracic surgery. Methods: In the study, 4012 consecutive patients were enrolled; among them, 71 patients (1.76%) had an unplanned return to the operating room. Age, sex, Charlson comorbidity index, induction treatments, type of the first operation, indication to readmission to the operating room and type of second operation, length of stay, complication after reoperation and outcomes were collected. Results: The mean age was 63.3 (SD: 13.0); there were 53 male patients (74.6%); the type of the first procedure was: lower lobectomy (11.3%), middle lobectomy (1.4%), upper lobectomy (22.5%), metastasectomy (5.6%), extrapleural pneumonectomy (4.2%), pneumonectomy (40.9%), pleural biopsy (5.6%) and other procedures (8.5%). Patients presenting complications after UROR had undergone a significantly longer first procedure (*p* < 0.02), had a longer length of stay (*p* < 0.001) and had higher post-operative mortality (*p* < 0.001). Conclusions: The patients experiencing UROR after elective oncologic thoracic surgery have significantly higher morbidity and mortality rates when compared to standard thoracic surgery. Bronchopleural fistula remains the most lethal complication in patients undergoing UROR.

## 1. Introduction

An unplanned return to the operating room (UROR) is defined as a readmission to the operating room because of a complication or an untoward outcome related to the initial surgery [1,2]. It has been widely used as an indicator of surgical care quality among hospitals [3,4,5] and is often advocated as a form of screening for assuring quality of care and identifying medical malpractice [6]. For these reasons, it is considered an effective outcome measure [7].

It has been previously reported that up to 70% of URORs are related to surgical complications, and patients readmitted to the operating room present significantly higher postoperative morbidity and mortality rates, longer in-hospital lengths of stay and increased needs of resources [8]. As a consequence, decreasing the UROR rate is paramount for the quality improvement of surgery [9]. A broad variability of URORs has been reported in the literature, ranging from 0.6% to 9% [7,10]; in fact, URORs are significantly influenced by many factors, such as type and setting of the operation [11], surgical technique, patients’ performance status and their comorbidities [12] as well as differences in coding practices between institutions [13]. URORs should, thus, be used as indicators of the quality of care only when specific criteria are met: the reoperation database needs to be extremely accurate, and this is provided only by unbiased, objective prospective registration [14,15,16]. In addition, preoperative patient comorbidities and performance status, as well as type and complexity of performed operations, should be correctly taken into consideration: in fact, it has been shown that the UROR rate was significantly higher in larger hospitals and referral centers, reflecting the very different case mix of patients treated at different institutions [17].

The value of historical surgical quality indicators—such as overall morbidity, severe morbidity and overall mortality—used for profiling hospital performances has recently been questioned because they can be significantly biased by several factors [18]; in fact, the mix of low caseload and low outcome rates decreases the power of many outcome indicators to discriminate the true quality differences among care providers, thus resulting in low reliability, similarly to power limitations in clinical trials [19].

While there has been a growing interest in the last years toward URORs as indicators of quality of care and instruments of pathway improvement in many surgical disciplines, only a few reports have addressed the role of URORs after oncologic thoracic surgery [9]. The aim of the present report is to evaluate the role of URORs after elective oncologic thoracic surgery in a high-volume, oncologic referral center, focusing on the risk factors and variables that influence the UROR rate to improve the quality of surgical care and minimize adverse events both during intraoperative and postoperative periods.

## 2. Materials and Methods

This was a single center, retrospective, observational study conducted in accordance with the Declaration of Helsinki as revised in 2013. The data were collected prospectively, entered into our institutional general thoracic database at the point of care, reviewed and double-checked, retrospectively. Written informed consent to undergo the procedure and for the use of clinical and imaging data for scientific or educational purposes, or both, was obtained from all patients before the operation. The study was approved by the medical ethics committee of our Institution (UID 2583 on 21 May 2021). Between January 2016 and December 2020, 4012 consecutive patients were operated on at our department because of a proven or suspected oncologic thoracic disease. Among them, 71 patients (1.76%) had an unplanned return to the operating room. An unplanned return to the operating room was defined as a readmission to the operating room within 90 days of the first operation because of a complication or an untoward outcome. A successful reoperation was defined as a procedure that was able to effectively control the complication causing the UROR.

Age, sex, Charlson comorbidity index, induction treatments, type of the first operation, indication to readmission to the operating room and type of second operation, length of stay, complications after reoperation and outcomes were collected for each patient.

### Statistical Method

Patient characteristics were summarized by the following: count and percent for categorical data; mean and Standard Deviation (SD) for normally distributed data or Interquartile Range (IQR) for non-normal data. These data were then tabulated by complication after reoperation and by reoperation during the same admission. Normality was checked with the Shapiro–Wilk test. Between group comparisons, the significance for categorical variables was tested by Fisher’s exact test, Wilcoxon or unpaired *t*-test for non-normal data and normally distributed data otherwise. All tests were two-tailed and considered significant at the 5% level. All analyses were completed by using STATA/MP 17.0 (Stata Corp. 2021. Stata Statistical Software: Release 17. College Station, TX, USA: Stata Corp LLC).

## 3. Results

The mean age was 63.3 (SD = 13.0); there were 53 male patients (74.6%); the mean preoperative Charlson comorbidity index was 5 (IQR = [4,6]); and the mean preoperative body mass index was 24.8 (SD = 4.3). The type of the first procedure was: lower lobectomy in 8 patients (11.3%), middle lobectomy in 1 patient (1.4%), upper lobectomy in 16 patients (22.5%), metastasectomy in 4 patients (5.6%), extrapleural pneumonectomy in 3 patients (4.2%), pneumonectomy in 29 patients (40.9%), pleural biopsy in 4 patients (5.6%) and other procedures in 6 patients (8.5%).

With regard to the number of URORs for each procedure in relation to all procedures of the same type performed in the study period, we observed the following: pneumonectomy 19.8%, extrapleural pneumonectomy 33.3%, upper lobectomy 8.6%, lower lobectomy 1.6%, metastasectomy 0.85%, pleural biopsy (VATS) 0.56% and middle lobectomy 1.25%.

The associated procedures were performed in 10 cases (14.1%); 48 procedures (67.6%) were performed on the right side, 22 (31%) on the left side and, in 1 case, we performed a combined median and right approach (1.4%). In 9 cases (12.7%), the patients previously received thoracic surgery on the same side, and in 17 cases (23.9%), the patients received induction treatments. Twelve patients (16.9%) received a minimally invasive approach, while 58 patients (81.7%) received an open approach at the time of the first procedure; 1 patient (1.4%) was submitted to chest drain positioning and—due to hemothorax—required readmission to the operating room to receive thoracoscopy for hemostasis. The first procedures had a mean duration of 192 min (IQR = [129,245]) (Table 1).

The indications for reoperation were: hemothorax in 39 patients (54.9%); bronchial fistula in 19 patients (26.8%); empyema in 3 patients (4.2%); prolonged air leaks not amenable of effective conservative treatment (Heimlich valve) in 3 patients (4.2%); wound dehiscence in 2 patients (2.9%); and other indications in 5 patients (7%). Thoracotomy for hemostasis was performed in 36 patients (50.7%); empyemectomy in 3 patients (4.2%); bronchial fistula suture in 9 patients (12.7%); thoracostomy in 8 patients (11.3%); thoracotomy for aerostasis in 2 patients (2.9%); thoracoscopy for aerostasis in 3 patients (4.2%); and other types of procedures in 10 patients (14.1%). The mean duration of the second surgical procedure was 107 min (IQR = [80,124]): A minimally invasive access was used in 5 cases (7%); an open approach was used in 65 cases (91.6%); and a chest tube positioning was used in 1 case (1.4%). Fifty-four patients (76.1%) were re-admitted to the operating room during the same hospital stay of the first procedure while 17 patients (23.9%)—who had already been discharged—needed a second admission to the hospital. The mean number of days between the first and second operation was 12 days (IQR = [1,15]). Sixty-one patients (85.9%) had an unplanned return to the operating room between postoperative day 0 and postoperative day 30; seven patients (9.9%) between postoperative day 31 and postoperative day 60; and three patients (4.2%) between postoperative day 61 and postoperative day 90.

The mean length of stay after the second operation was 30 days (IQR = [9,48]). A reoperation was successful in 57 patients (80.2%) (Table 2).

Among these 57 patients, only 2 died (3.5%); among the 14 patients (19.8%) who had an unsuccessful reoperation, 7 (50%) died (*p* < 0.001); in 6 out of 7 deceased patients (85.7%) the indication for a UROR was bronchopleural fistula (Table 3).

Complications after operations were reported in 32 patients (45.1%); atrial fibrillation was observed in 6 patients (8.5%); bronchial fistula in 6 patients (8.5%); dysphonia in 5 patients (7.0%); prolonged air leaks in 2 patients (2.8%); respiratory failure and intubation in 10 patients (14.1%); and other type of complications in 3 patients (4.2%). Nine patients (12.7%) died after the second operation (Table 4). In the study period, we observed 75 unplanned intensive care unit (ICU) admissions (1.86%).

Patients who had an unplanned return to the OR during the same hospital admission had a significantly shorter length of stay (mean 21 vs. 58 days, *p* < 0.001) (Table 5).

Patients presenting with complications after a UROR had been submitted to a significantly longer first procedure (mean 223 min vs. 167 min, *p* = 0.02), had a significantly longer length of stay (41.4 days vs. 20.5 *p* < 0.001) and had a significantly higher post-operative mortality (9 pts vs. 0 pts, *p* < 0.001) (Table 6).

## 4. Discussion

It has been reported that a large proportion of adverse events in hospitalized patients occurs in surgical patients [3]; about half of these events have been described as “preventable” [20]. For this reason, there has been growing interest toward indicators that could effectively quantify the incidence of postoperative adverse events and measure systematically clinical outcomes. Although postoperative morbidity and mortality rates represent the most frequently used parameters to assess the quality of surgical care, it has been correctly observed that mortality is usually rare for most procedures, and non-lethal complications are commonly strictly related to specific procedures [3], thus not allowing a comprehensive standardization of the value of surgical care.

URORs are more common than postoperative mortality for most surgical procedures and—being reported for almost every kind of procedure—they are widely applicable as a quality indicator; in addition, URORs are basically non-discretionary events in the sense that a readmission to the operating room is planned only when patients really need readmission; on the contrary, other measures may be more easily biased by a single operator’s evaluation [3]. For all these reasons, URORs have emerged as an effective instrument for quality evaluation and medical malpractice assessment [21,22,23,24,25,26,27,28].

URORs within 24 h of the first procedure form a subgroup that may offer additional information: It has been reported that the most frequent indication is bleeding, as expected in almost every type of surgery. The main risk factors for readmission to the operating room within the first postoperative day have been found to be a history of liver disease, smoking, reduced preoperative platelet count and preoperative administration of anticoagulant or antiplatelet drugs [29]. Similarly, in our experience, all readmissions to the operating room on postoperative day 0 or 1 (18 patients) were due to hemothorax after open surgery, and this further explains why we preferred an open UROR approach rather than a minimally invasive one. On the other hand, we tried a UROR VATS approach in cases of previous minimally invasive surgery in patients without any hemodynamic instability.

We further studied this subgroup of patients, but we did not find any significant risk factor related to postoperative haemothorax (liver function, antiplatelet and anticoagulant drugs assumption). In the vast majority of cases, we did not find a clear cause of bleeding during UROR surgery; on the contrary—only in a small proportion of cases—we were able to identify intercostal or bronchial arteries as an active source of bleeding.

The vast majority of our patients experiencing UROR were, in fact, re-operated on during the first 30 post-operative days (85.9%). Only a small percentage of patients experienced UROR between post-operative days 31 and 60 (9.9%) and between post-operative days 61 and 90 (4.2%), confirming that the longer the post-operative course, the lower the chance of UROR after elective oncologic thoracic surgery. In our experience, post-operative morbidity and mortality rates after UROR are definitely higher when compared to standard elective oncologic thoracic surgery, which were 45% and 12.6%, respectively, versus 36.2% and 1.3% in the non-UROR group. Moreover, the mortality rate in patients with complications after UROR increased to 28.1% with respiratory failure needing oro-tracheal intubation being the most lethal complication. This is probably due not only to technical issues strictly related to the surgical treatment of the complication itself but also to the global management of patients. In fact, these patients—despite all protective measures employed by the anesthesiologist during UROR—easily developed transfusion-related acute lung injury (TRALI), acute respiratory distress syndrome (ARDS) or acute lung injury (ALI) that significantly impacted the post-operative course. It is worth underlining that patients with complications after UROR received a significantly longer first operation when compared to uncomplicated UROR (223 vs. 167 min), while no difference was observed in terms of the duration of the second operation. A longer duration of the surgical procedure, in fact, may itself represent a risk factor for UROR; however, it is an indirect indicator of the complexity and difficulty of surgery, thus clearly explaining why a longer-lasting procedure correlates with a higher post-UROR morbidity rate.

It is interesting to underline that—in our experience—none of the variables related to the patients showed a significant impact on post UROR complications: Age, sex, preoperative body mass index, Charlson comorbidity index, side, type of access, induction treatments, previous chest surgery and associated procedures did not play any role in terms of complications, thus suggesting that the most influencing factor is the first procedure itself rather than the patient’s clinical status (Appendix A).

Our findings showed that patients experiencing UROR during the same hospital admission have a significantly shorter length of stay when compared to patients requiring a second admission due to UROR; this is probably due to the type of complication conditioning UROR. As reported before, hemothorax is the most frequent cause of UROR shortly after surgery, and when successfully managed, it does not significantly impact the total length of stay, which is almost similar to that of patients with an uncomplicated post-operative standard course. On the other hand, late bronchial healing problems are the most frequent causes of UROR requiring readmission after previous discharge following the first procedure; they need more complex treatments (e.g., thoracostomy) causing a longer total length of stay.

In our experience, a successful reoperation—defined as a procedure that is able to effectively control the complication causing UROR—showed a limited mortality of 3.5%. On the contrary, an unsuccessful reoperation disclosed a significantly higher postoperative mortality of 50%, thus underlining the pivotal role of an effective redo procedure in the global management of surgical complications. Bronchopleural fistula was the indication with the higher postoperative mortality rate, once again showing how dangerous this complication can be even when promptly approached [30,31].

## 5. Conclusions

UROR is an effective and reliable quality indicator in oncologic thoracic surgical care. Patients experiencing UROR after elective oncologic thoracic surgery have a higher morbidity and mortality rates when compared to standard thoracic surgery. Patients presenting with complications after UROR had been submitted to a significantly longer first procedure, had a significantly longer length of stay and a higher post-operative mortality. Bronchopleural fistula remains the most lethal complications in patients undergoing UROR.

## Figures and Tables

**Table 1 cancers-14-02064-t001:** Patients’ demography, treatments and procedures summary statistics, N = 71.

Variables		Statistics *^a^*
Age at Surgery, years		63.3 (13.0)
BMI		24.8 (4.3)
Charlson Comorbidity Index		5.0 (4,6)
Male Gender		53 (74.6)
Type of Procedure	Pneumonectomy	29 (40.9)
	Upper Lobectomy	16 (22.5)
	Lower Lobectomy	8 (11.3)
	Metastasectomy	4 (5.6)
	Pleural Biopsy (vats)	4 (5.6)
	Extrapleural Pneumonectomy	3 (4.2)
	Middle Lobectomy	1 (1.4)
	Other	6 (8.5)
Side	Right	48 (67.6)
	Left Median + Right	22 (31.0) 1 (1.4)
Associated Procedures		10 (14.1)
Previous Chest Surgery		9 (12.7)
Induction Treatments		17 (23.9)
Access and Duration	Open	58 (81.7)
	Minimally Invasive	12 (16.9)
	Not Available	1 (1.4)
	Duration (minutes)	192 (129,245)

*^a^* Mean (SD) for age and BMI, mean (IQR) for Charlson comorbidity index and duration, N (%) otherwise. SD = Standard Deviation; IQR = Interquartile Range.

**Table 2 cancers-14-02064-t002:** Reoperations summary statistics, N = 71.

Variables		Statistics *^a^*
Type of Reoperation	Thoracotomy for Haemostasis	36 (50.7)
	Bronchial Fistula Suture	9 (12.7)
	Thoracostomy	8 (11.3)
	Empiemectomy	3 (4.2)
	Thoracoscopy for Aerostasis	3 (4.2)
	Thoracotomy for aerostasis	2 (2.9)
	Other	10 (14.1)
Indications for Reoperation	Heamothorax	39 (54.9)
	Bronchial Fistula	19 (26.8)
	Empyema	3 (4.2)
	Prolonged Air Leaks	3 (4.2)
	Wound Dehiscence	2 (2.9)
	Other	5 (7.0)
Access and Duration	Open	65 (91.6)
	Minimally Invasive	5 (7.0)
	Not Available	1 (1.4)
	Duration (minutes)	107 (80,124)
Reoperation during the Same Admission		54 (76.1)
Days between Operations		12 (1,15)
Reoperation on Postop Day	0–30	61 (85.9)
	31–60	7 (9.9)
	61–90	3 (4.2)
Length of Stay after Reoperation (days)		30 (9,48)
Successful Operation		57 (80.3)

*^a^* Mean (IQR) for duration, days between operation and length of stay, N (%) otherwise. SD = Standard Deviation; IQR = Interquartile Range.

**Table 3 cancers-14-02064-t003:** Frequency distribution of deaths by reoperation outcome.

	Successful Reoperation	
Post Reintervention Death	No N = 14	Yes N = 57
No	7 (50.0)	55 (88.7)
Yes	7 (50.0)	2 (3.5)

Statistics are: N (column %); *p* < 0.001.

**Table 4 cancers-14-02064-t004:** Complications and deaths after reoperation distribution frequency, N = 71.

Type of Complication	N (%)
Respiratory Failure and Intubation	10 (14.1)
Atrial Fibrillation	6 (8.5)
Bronchial Fistula	6 (8.5)
Dysfonia	5 (7.0)
Prolonged Air Leaks	2 (2.8)
Other	3 (4.2)
Any Complication	32 (45.1)
Deaths	9 (12.7)

**Table 5 cancers-14-02064-t005:** Patients’ characteristics and treatments by reoperation during the same admission summary statistics.

		Same Admissiom		
Characteristic		No N = 17	Yes N = 54	*p*-Value
Age (years)		63.0 (17.7)	63.4 (11.3)	0.51
BMI (kg/m^2^)		25.9 (3.7)	24.5 (4.5)	0.28
Duration of First Operation (min)		180 (138,252)	196 (119,242)	0.75
Duration of 2nd Operation (min)		111 (88,120)	106 (75,124)	0.32
Days to 2nd Intervention		34 (14,45)	6 (1,5)	<0.001
Length of Stay (days)		58 (27,69)	21 (8,31)	<0.001
Charlson Comorbidity Index		5.3 (3.0,7.0)	5.0 (4.0,6.0)	0.53
Sex	Female	5 (29.4)	13 (24.1)	
	Male	12 (70.6)	41 (75.9)	0.75
Side	Left	3 (17.7)	19 (35.2)	
	Right	13 (76.5)	35 (64.8)	
	Right + Median 1 (5.9)		0	0.14
Access	Open	16 (94.1)	49 (90.7)	
	Minimally Invasive	0	5 (9.3)	
	Not Available	1 (5.9)	0	0.21
Induction Treatments		3 (17.7)	14 (25.,9)	0.74
Associate Procedures		1 (5.9)	9 (16.7)	0.43
Previous Surgery		2 (11.8)	7 (13.0)	1.00
Re-Surgery Complication		8 (47.1)	24 (44.4)	1.00
Deaths		4 (23.5)	5 (9.3)	0.20

Statistics for continuous variables are mean (SD) for age and BMI, mean (IQR) otherwise; N (%) for categorical variables; SD = Standard Deviation; IQR = Interquartile Range.

**Table 6 cancers-14-02064-t006:** Patients characteristics and treatments by complication after reoperation summary statistics.

		Complication		
Characteristic		No N = 39	Yes N = 32	*p*-Value
Age (years)		61.6 (14.2)	65.3 (11.2)	0.14
BMI (kg/m^2^)		24.7 (4.9)	25.0 (3.6)	0.45
Duration of First Operation (min)		167 (105,208)	223 (150,268)	0.02
Duration of 2nd Operation (min)		106 (79,124)	109 (83,125)	0.78
Days to 2nd Intervention		9.9 (1,11)	15.4 (2,23)	0.13
Length of Stay (days)		20.5 (8,26)	41.4 (14,50)	<0.001
Charlson Comorbidity Index		5.1 (3.0,6.0)	5.0 (4.0,5.5)	0.62
Sex	Female	9 (23.1)	9 (28.1)	
	Male	30 (76.9)	23 (71.9)	0.78
Side	Left	14 (35.9)	8 (25.0)	
	Right	25 (64.1)	23 (71.9)	
	Right + Median	0	1 (3.1)	0.31
Access	Open	34 (87.2)	31 (96.9)	
	Minimally Invasive	4 (10.3)	1 (3.1)	
	Not Available	1 (2.6)	0	0.37
Induction Treatments		6 (15.4)	11 (34.4)	0.09
Associate Procedures		3 (7.7)	7 (21.9)	0.17
Previous Surgery		5 (12.8)	4 (12.5)	1.00
Deaths		0	9 (28.1)	<0.001

Statistics for continuous variables are mean (SD) for age and BMI, mean (IQR) otherwise; N (%) for categorical variables; SD = Standard Deviation; IQR = Interquartile Range.

## Data Availability

Available upon request.

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
