# Peer review of "Unplanned Return to the Operating Room after Elective Oncologic Thoracic Surgery: A Further Quality Indicator in Surgical Oncology"

_cancers, 2022, doi:10.3390/cancers14092064_

Round 1

Reviewer 1 Report

Dear authors,
I was allowed to review your manuscript. Congratulations on this interesting work, which explores an important topic. I think this work is interesting for the readers. However, I have the following questions or comments on the manuscript:
In the results, you focus almost entirely on the study group (n= 71) with reoperations. It would be interesting to know what the percentage of reoperations is in relation to the individual primary operations. In other words, how many re-operations were performed after lobectomy, pneumectomy, EPP, metastasectomy, etc.? What is the risk after VATS lobectomy vs. thoracotomy with lobectomy?
In my opinion, this should be added to the manuscript.
A comparison of no UROR (n= 3941) vs. UROR (n= 71) would therefore be interesting. These data need to be shown. From this, one could identify the risk factors or a risk constellation. This is crucial to confirm your conclusions that patients with UROR have a higher morbidity/mortality compared to normal thoracic patients. So far, these results are lacking...
Overall, the manuscript should also be a bit more focused, as there is a lot of data shown, which makes it a bit confusing to read. 

Author Response

Reviewer 1

Dear authors,
I was allowed to review your manuscript. Congratulations on this interesting work, which explores an important topic. I think this work is interesting for the readers. However, I have the following questions or comments on the manuscript:
In the results, you focus almost entirely on the study group (n= 71) with reoperations. It would be interesting to know what the percentage of reoperations is in relation to the individual primary operations. In other words, how many re-operations were performed after lobectomy, pneumectomy, EPP, metastasectomy, etc.? What is the risk after VATS lobectomy vs. thoracotomy with lobectomy?
In my opinion, this should be added to the manuscript.

 Pneumonectomy  19.8%

 Extrapleural pneumonectomy 33.3%

Upper lobectomy 8.6%

Lower lobectomy 1.6%

Metastasectomy 0.85%

Pleural biopsy (VATS) 0.56%

 Middle lobectomy 1.25%

Revised version #1 page 3  lines 115 - 118

Although not statistically significant, minimally invasive lobectomy (VATS and RATS lobectomy) had a lower UROR rate when compared to thoracotomic lobectomy; anyway, given the fact that we reserved minimally invasive lobectomy only to very selected patients (early stage, no induction, no re do surgery) the two cohorts of patients are not homogeneous and a correct comparison can’ t be performed. being the vast majority of UROR patients in the open approach group

Revised Version #1 table 5 and Table 6

A comparison of no UROR (n= 3941) vs. UROR (n= 71) would therefore be interesting. These data need to be shown. From this, one could identify the risk factors or a risk constellation. This is crucial to confirm your conclusions that patients with UROR have a higher morbidity/mortality compared to normal thoracic patients. So far, these results are lacking...

No-UROR (3941)  morbidity and mortality were  36.2% and 1.3% respectively and both were significantly lower than those of the  UROR group where morbidity and mortality were  45.1%  and  12.7 % respectively.

Revised Version #1  Page 11, lines 32,33

Overall, the manuscript should also be a bit more focused, as there is a lot of data shown, which makes it a bit confusing to read. 

We try to better focus on the major topics in the new discussion section

Reviewer 2 Report

The authors declared that unplanned return to the operating room (UROR) after elective oncologic thoracic surgery should be an effective and reliable quality indicator in oncological thoracic surgical care. The conclusions are reasonable; the early reoperation is due to heamothorax and the late one is due to bronchopleural fistula, which remains the most lethal complication.

However, this analysis could not assess the other lethal medical malpractice such as ARDS and severe respiratory failure, which could be evaluated by the incidence of unplanned critical care admission.

Therefore, I hope the author could note the clinical indicators on mortality and morbidity (unplanned critical care admission) in their institute during the study period.

Author Response

Reviewer 2

The authors declared that unplanned return to the operating room (UROR) after elective oncologic thoracic surgery should be an effective and reliable quality indicator in oncological thoracic surgical care. The conclusions are reasonable; the early reoperation is due to heamothorax and the late one is due to bronchopleural fistula, which remains the most lethal complication.

However, this analysis could not assess the other lethal medical malpractice such as ARDS and severe respiratory failure, which could be evaluated by the incidence of unplanned critical care admission.

Therefore, I hope the author could note the clinical indicators on mortality and morbidity (unplanned critical care admission) in their institute during the study period.

In the study period we had 75 unplanned ICU admissions (1.86%) whose mortality was 24%. Acute respiratory failure requiring re do orotracheal intubation was the most common indications for unplanned ICU admission.

Revised version #1, page 5, lines 127 -128.

Reviewer 3 Report

This is an interesting study and the authors have collected a unique dataset using cutting edge methodology. The paper is generally well written and structured. 

The topic is very interesting because it shifts our focus on what may be the operative complications, analyzing what happens after the same complication that we all know.

It might be interesting, starting from this point, to dwell on the most common comolences that came out of this study to understand if they were due to specific repeated causes, in order to minimize adverse situations in the future.

  • Line 127-128 Define the time of prolnonged air leaking taht you have decide to came back to the surgery room . What kind of conservative traetment have you used precedntly and for how long?

  • Have you seen different for UROR based on the different kind of fisrt  approach? Open versus minimally invasive ?­

  • For the second intervention you have preferred  use a open  approach in the majority of cases.  It could be explain because of the emergency of the situation [ major bleeding and unstable vital sign ]?

  • Hem thorax represents FOR 54.9% the most common event that could which may require re-evaluation in the operating room. In your datas have you studied this group of patients? it could depend on an incorrect suspension of anti-coagulant or anti-aging drugs in the pre or post operative time of surgery ? It could be possibile to identify the surgical incident that have the surgical accident which caused the most expensive post-surgical bleeding, also in order to be able to identify a series of cases to create a

    a collective memory of adverse events

Author Response

Reviewer 3

This is an interesting study and the authors have collected a unique dataset using cutting edge methodology. The paper is generally well written and structured. 

The topic is very interesting because it shifts our focus on what may be the operative complications, analyzing what happens after the same complication that we all know.

It might be interesting, starting from this point, to dwell on the most common comolences that came out of this study to understand if they were due to specific repeated causes, in order to minimize adverse situations in the future.

  • Line 127-128 Define the time of prolnonged air leaking taht you have decide to came back to the surgery room . What kind of conservative traetment have you used precedntly and for how long?

Although standard definition of prolonged air leaks is “air coming out of the remaining lung tissue until the fourth, fifth or seventh postoperative day, according to different classifications”, in this paper we referred to prolonged air leaks not amenable of conservative treatments.

As first step we tried to use Hemlich valve after 6 days of prolonged air leaks; if the valve was not tolerated (major pneumothorax, subcutaneous emphysema, dyspnoea) we than decided for re-operation for surgical management of aerostasis.

Revised version #1, page 4, lines 164 -165.

  • Have you seen different for UROR based on the different kind of fisrt  approach? Open versus minimally invasive ?­

Although not statistically significant, minimally invasive lobectomy (VATS and RATS lobectomy) had a lower UROR rate when compared to thoracotomic lobectomy; anyway, given the fact that we reserved minimally invasive lobectomy only to very selected patients (early stage, no induction, no re do surgery) the two cohorts of patients are not homogeneous and a correct comparison can’ t be performed, being the vast majority of UROR patients in the open approach group.

              Revised Version table 5 and Table 6

  • For the second intervention you have preferred  use a open  approach in the majority of cases.  It could be explain because of the emergency of the situation [ major bleeding and unstable vital sign ]?

In the vast majority of case we used open approach because indication was haemothorax in post operative day 0 or 1 after thoracotomic surgery, so VATS approach was skipped; on the other side, we tried UROR VATS approach in case of previous minimally invasive surgery in patients without any haemodinamic instability.

             Revised version #1, page 11 lines 23 -26

  • Hemothorax represents FOR 54.9% the most common event that could which may require re-evaluation in the operating room. In your datas have you studied this group of patients? it could depend on an incorrect suspension of anti-coagulant or anti-aging drugs in the pre or post operative time of surgery ?

We further investigated  this subgroup of pateints focusing on comorbidities, liver function, antiplatelets and anticoagulant drugs but we did not find any significant risk factor  related to post operative haemothorax

Revised version #1, page 11 lines 28-30

  • It could be possibile to identify the surgical incident that have the surgical accident which caused the most expensive post-surgical bleeding, also in order to be able to identify a series of cases to create a a collective memory of adverse events

In the vast majority of cases we did not find a clear cause of bleeding during UROR surgery; on the contrary - only in a small proportion of case - we were able to identify intercostal or bronchial arteries as an active source of bleeding.

              Revised version #1, page 11 lines 30 - 32

Round 2

Reviewer 1 Report

Dear authors,

thank you very much for the revised version of your manuscript. My suggestions have been adequately implemented.